# Retinal Cell Protection in Ocular Excitotoxicity Diseases. Possible Alternatives Offered by Microparticulate Drug Delivery Systems and Future Prospects

**DOI:** 10.3390/pharmaceutics12020094

**Published:** 2020-01-24

**Authors:** Javier Rodríguez Villanueva, Jorge Martín Esteban, Laura J. Rodríguez Villanueva

**Affiliations:** 1Human resources for I+D+i Department, National Institute for Agricultural and Food Research and Technology, Ctra. de la Coruña (Autovía A6) Km. 7.5, 28040 Madrid, Spain; 2University of Alcalá, Ctra. de Madrid-Barcelona (Autovía A2) Km. 33,600, 28805 Alcalá de Henares, Madrid, Spain; jorge.martin@edu.uah.es (J.M.E.); julia.rodriguez@edu.uah.es (L.J.R.V.)

**Keywords:** excitotoxicity, neurodegeneration, retina, microparticles, controlled drug release

## Abstract

Excitotoxicity seems to play a critical role in ocular neurodegeneration. Excess-glutamate-mediated retinal ganglion cells death is the principal cause of cell loss. Uncontrolled glutamate in the synapsis has significant implications in the pathogenesis of neurodegenerative disorders. The exploitation of various approaches of controlled release systems enhances the pharmacokinetic and pharmacodynamic activity of drugs. In particular, microparticles are secure, can maintain therapeutic drug concentrations in the eye for prolonged periods, and make intimate contact by improving drug bioavailability. According to the promising results reported, possible new investigations will focus intense attention on microparticulate formulations and can be expected to open the field to new alternatives for doctors, as currently required by patients.

## 1. Introduction

Glutamate plays an important function in the regulation of relevant neurophysiological processes. For example, it has been proposed as a molecular substrate for learning and memory [1]. However, long-term overactivation of excess glutamate receptors in the central nervous system is associated with neuronal cell death. In the eye, an outward extension to interact with the world around us, controlled by the autonomic and central nervous systems, this happens in exactly the same way. It is released by photoreceptors, bipolar and ganglion cells and is responsible for the transmission of the light signal [2]. Examples of the participation of glutamate in other eye functions are related with synaptic transmission and plasticity, the transcriptional control of the glutathione biosynthesis and the maintenance of the cellular redox balance [3]. Glutamate homeostasis in the eye is controlled among others by macroglial cells (such as astrocytes and Müller cells) and the microglia. Microglia phagocytes monitor the environment and are rapidly alerted by a variety of injurious signal inputs, triggered by either genetic or environmental factors, as the result of external damage from ocular infections or due to cellular malfunctions. Their activation causes the secretion of neuroactive molecules, termed “gliotransmitters” affecting neurotransmission [4]. These include, among others, neurotransmitters (ATP, glutamate), eicosanoids, lactate, and the cytokine tumor necrosis factor α, factors that modulate astrocytes and Müller cells, thus ensuring proper functioning of the healthy retina [5].

When a rapid return to normality is not achieved, uncontrolled high concentrations of glutamate in synaptic and extrasynaptic locations lead to excitotoxicity, a downstream mechanism that activates cytotoxic cascades culminating in neuronal damage and/or death [6]. This excess glutamate (and related compounds that mimics its action) induced excitotoxicity acts primarily through *N*-methyl-d-aspartate acid (NMDA) receptors (NMDARs) [7], which, with kainite receptors (KARs) and α-amino-3-hydroxy-5-methyl-4-isoxazolepropionic acid receptors (AMPARs), is one of the three-ionotropic glutamate receptor subtypes expressed abundantly in inner retinal cells, making them particularly susceptible to excitotoxicity [8]. NMDARs are usually composed of two obligatory GluN1 subunits and two regulatory GluN2 subunits (GluN2A, 2B, 2C, 2D) [9]. The mechanism of action is related with a perturbation of Na^+^/K^+^ homeostasis and an excessive calcium influx (Ca^2+^ overload), activation of extracellular signal regulated kinases 1/2 (ERK 1/2) [10] and increased release of reactive oxygen species (ROS), oxidative stress, and cytochrome c (mitochondrial dysfunction), resulting in apoptotic cell death (Figure 1). Intravitreal injection of excess glutamate has elicited loss of retinal ganglion cells (RGC) [11], the output neurons of the retina that connect the eye to the brain [12]. RGC collect and integrate visual information from second-order neurons and then transmit electrical impulses from the retina to the brain [8]. Furthermore, it reduces thickness of the retinal nerve fiber layer and attenuates the amplitude of the a- and b-wave components in pattern electroretinogram [13]. Another value, implicit time, is also altered in a- and b-waves, and oscillatory potentials are drastically affected by glutamate disturbances, indicating a neuronal dysfunction [5].

Excitotoxicity has been linked to the pathogenesis of several serious degenerative ocular diseases and injuries [9]. Apart from glaucoma, retinitis pigmentosa [14], diabetic retinopathy [15], retinal ischemia [16], age-related macular degeneration (AMD) [17] or Leber hereditary optic neuropathy are characterized by progressive retinal neurodegeneration. These disorders include cell stress and inflammatory response, at least in part due to increased glutamate levels [18], what ends in retinal remodeling. RGC and photoreceptors death by apoptosis is the characteristic of most degenerative ocular disorders [19].

Among them, particular emphasis should be placed on glaucoma. This disease remains the main reason for irreversible visual loss worldwide and the second most common in many developed countries [20,21]. It is estimated that this disease affected approximately 70 million people in 2014 [22] and the prevalence of this disease is expected to grow as the population ages, affecting more than 80 million people worldwide by 2020, with at least 6 to 8 million of them becoming bilaterally blind. It is a neurodegenerative disease of the inner retina and optic nerve characterized by morphological change of the optic nerve head and progressive degeneration of RGC and their axons. Glaucoma is subdivided into three main groups: open-angle glaucoma, closed-angle glaucoma, and secondary glaucoma. Furthermore, open-angle glaucoma is subdivided into high-pressure glaucoma and low-pressure glaucoma [23]. In closed-angle glaucoma, the peripheral iris blocks the anterior chamber angle by apposition or synechiae, preventing the drainage of the aqueous humor. In the case of open-angle glaucoma, the anterior chamber angle is open but the aqueous humor drainage through the trabecular meshwork is almost shut (especially in the juxtacanalicular tissue regions). Primary glaucoma is not associated with pre-existing diseases whereas secondary glaucoma is a consequence of another ocular or systemic disease, trauma, or from drug adverse effects [24].

While elevated intraocular pressure (IOP) is considered the major risk factor for causing optic nerve damage, ocular ischemia and subsequent remodelation (among others by endotheline-1 release), and lowering IOP is well established as a glaucoma pharmacological or surgical treatment, the precise causal mechanisms of glaucoma are not fully understood. In some patients, IOP reduction alone is not adequate and they continue to lose vision despite well-controlled IOPs [25]. Recent research has shown that it is the death of RGC that manifests visual field deficits. For this reason, it is critical to develop treatment that actively prevents the death of RGC, which are at risk in this condition [26]. Damage initiates by uncontrolled inflammation and excitotoxicity in response to ischemic injury, with lack of visual symptoms until substantial damage has occurred to the RGC and primary symptom of glaucoma, progressive loss of the visual field, appears. NMDA receptor antagonists have been used to treat neurodegenerative diseases in clinics. However, the universality of the glutamic acid neurotransmitter system makes the glutamic acid receptor blockers inefficient and unsafe in clinical experiments [27].

Until now, there is no consensus on a clinical approach to treat glaucoma by directly targeting the neural tissue to protect them from degeneration [28]. This is partly due to the numerous numbers of different substances that have been suggested as candidates for neuroprotective therapy or neuroprotection. This is the term used for therapies that are independent of IOP lowering [21] and that protect the RGC by inhibiting various mechanisms of RGC degeneration and apoptosis, or promoting their survival [29]. The authors of [30] and [27] proposed that, among others, RGC survival can be achieved by blocking apoptosis (as, i.e., caspase and semaphorin inhibitors do), preventing glutamate induced RGC excitotoxicity (using NMDA receptor antagonists such as memantine), acting on mitochondrial injury (i.e., through the iron ion metabolism system as deferoxamine), administering various “neurotrophins” (i.e., glial cell line-derived neurotrophic factor, ciliary neurotrophic factor or brain-derived neurotrophic factor), free radical capturing elements (such as carnitine, carnosine, co-enzyme Q10, omega-3 fatty acids, flavonoids as hesperidin or vitamin B12) or calcium channel blockers (i.e., “-dipine” drugs or possibly anti-inflammatory compounds with a free carboxyl groups such as ketorolac, ibuprofen or dexamethasone [31]).

Nowadays, it is known that preservation of adequate protein folding, mitochondrial function, inflammation process and scavenging pathways directly avoids programmed cell death and irreversible damage of the neuronal cells [32]. Also, inhibition of retinal microglial and Müller glia activation via sigma-1 receptor agonism with neurosteroids (e.g., dehydroepiandrosterone, that has anti-inflammatory functions as well) has been described recently as an innovative and viable strategy [28]. However, not only should the pathway and the mechanism be effective, but also the drug should access the target site and maintain availability of the active agent for prolonged periods of time.

Solid ophthalmic devices, contact lenses, viscous liquids, gels, suspensions, colloidal systems (nanoparticles and nanosuspension), matrix system (ocular inserts, minitablets and collagen shields) liposomes, dendrimers, solid lipid nanoparticles, niosomes, and microparticles (the intraocular drug delivery systems or IODDS) have been developed with varying success. They emerge as an interesting alternative to repeated intravitreal injections of active substances in solution [33]. The administration of IODDS not only avoids serious complications related to frequent repeated injections, i.e., endophthalmitis and retinal detachment [34], but also reduces the risk of initial toxicity, due to high local drug concentrations typically occurring when they are administered in bolus [14]. Because of their micro- and nano-sized dimensions, the use of particulate systems has been explored as an appropriate alternative to conventional options in ophthalmology. They offer the possibility to enhance delivery and transport of drugs across ocular tissues [35].

However, until now, nano-sized systems have hardly been able to maintain stable drug concentration levels due to their small size they are cleared quickly through the conjunctival, sclerotic, and other periocular circulation systems [36]. Microparticles (MP), even so, are able to release an active substance for longer periods of time (up to 6 months) compared to nanoparticles. Generally, when they are elaborated for ophthalmic use, they must be extremely small-sized to avoid ocular damage due to abrasion and irritation while also providing easy injectability. Particles of a diameter less than 100 μm are considered suitable for intravitreal administration [37]. When MP are in a range between 2 to 40 μm, they can be injected through devices that include a needle (30–32 G) in a minimally invasive intervention. Among their possibilities there must be highlighted that they are good candidates to be used in personalized medicine as different amounts of particles can be administered depending on patient needs [38] and drug dosage could also be diminished to optimal, leading to a reduction of the possible side effect [39]. After administration, microparticles are not expected to move as they have the tendency to aggregate several days and remains in the vitreous cavity. Aggregation have been suggested, by some authors cited in [38], as a phenomenon that contributes to microparticles’ implant behavior, what would led to some benefits as predictable kinetics. MP prepared with polylactic-co-glycolic acid (PLGA), a biodegradable polymer that is gradually converted into CO_2_ and water in vivo by ocular tissues, revealed safe, stable, biocompatible, without intrinsic immunogenicity and relatively easy to produce on a large scale [40]. It fulfills the basic requirements that an ideal material for a vector should possess: compatibility with the host tissue, no immunological reaction, minimal damage at the injection site and controlled release of the drug.

In this review, we present the latest and most promising scientific communications on the field of ocular neuroprotection, focusing on those where microparticles are employed as drug controlled release systems. Emphasis, when possible, is made on their security profile and the drug release period these systems achieve as a potential advantage among intravitreal injections. This work is divided in three chapters. In the first one the intervention over excess glutamate is screened. Then, the use of neuroprotective therapies is discussed. In this case, the chapter is divided in two different parts. First, one for neurotrophic factors and then a second, where the possibility of using antiapoptotic compounds is analyzed. Finally, the approach of maintaining the retinal structure integrity as an alternative is considered.

## 2. Intervention over Excess Glutamate

If excitotoxicity is caused primarily by overactivation of NMDA receptors, the first idea proposed as a pharmacological approach for the treatment of degenerative diseases has been NMDA receptor antagonism where excitotoxicity is involved. However, NMDA receptors’ complete blockade, as has been previously mention, implies intolerable side effects [8]. Another possibility, then, is reducing the level of glutamate in the synapsis. To this end, scientists know that one of the important functions of Müller cells is the regulation of synaptic activity through preserving low concentration of glutamate via activation of the P2 × 7 receptor [41] or upregulating high-affinity carriers such as the excitatory amino acid transporter 1 (EAAT1) or 2 (EAAT2) [42] and the glutamate/aspartate transporter (GLAST) [20]. In this last case, their activation by an α_2_ agonist may contribute to the neuroprotective effects in RGC by decreasing synaptic glutamate levels. Brimonidine is a selective alpha 2-adrenergic agonist that has demonstrated the ability to lower intraocular pressure and protect retinal ganglion cells against glutamate excitotoxicity. Its mechanisms of action include inhibition of glutamate release, upregulation of brain-derived neurotrophic factor expression, regulation of cytosolic Ca^2+^ signaling, modulation of *N*-methyl-d-aspartate receptor through *N*-methyl-d-aspartate receptors 1 and 2A protein expression, reduction of aqueous humor production by the ciliary body and increased clearance through either the trabecular meshwork into the episcleral veins or the uveoscleral outflow pathway into the suprachoroidal space [43,44,45].

In-vivo studies carried out by [46] demonstrated that brimonidine in intravitreal application (3.6 nmol) protects retinal ganglion cells in a rabbit retinal NMDA excitotoxicity model. Ocular excitotoxicity severity correlates well with RGC disturbance. A study of viable RGC, their morphology and functionality can give yield information about ocular diseases progression and the results of therapeutical interventions. In line with [12], most of the current research investigating the pathological mechanisms underlying RGC death or evaluating novel therapeutics is conducted in genetic or experimental mouse models of glaucomatous optic neuropathies. All these models share their dependence on RGC quantification to follow the course of glaucomatous degeneration. The multiple methods that can be used to visualize RGCs in naive and experimentally manipulated rodent retinas includes basic histological [47], Nissl staining, neuronal immunostainings, retrograde tracers (such as fluorogold, dextran tetramethylrhodamine, or DiI into areas of the brain that are targeted by RGCs or by exposure of an axotomized optic nerve to these dyes [48]) or specific RGCs immunostainings (highlighting Brn3a, a POU domain class 4 transcription factor able to label their nucleus). After that, RGC counting can be done on retinal sections or in retinal flatmounts. In both methods, small frames are selected from the central, mid-peripheral and peripheral retina and RGC density is averaged and often extrapolated for the entire retina as a percentage versus control. As a result, the most common problems are related with inter- and intra-observer variation. Furthermore, if it is done manually, the technique proves to be laborious. In animal models of retinal degeneration, a- and b-wave amplitudes provide key information about disease-associated functional changes in the retina, but also about morphological alterations occurring during degenerative processes, including the progressive loss of photoreceptors and synaptic connectivity impairment [5]. In spite of this, most current animal models display rapid RGC degeneration incident, what is atypical of most human glaucomas and cannot be assumed to accurately reflect the disease process but up to date they are the most reliable preclinical models to develop pharmaceutics.

Later, [45] corroborated this observation and demonstrated that 0.2 mg brimonidine/day intravitreally injected to female Sprague-Dawley rats were enough to protect RGC specifically against mitochondrial dysfunction, the main rescue mechanism activated, induced by glutamate excitotoxicity and/or oxidative stress in ischemic retina. This small amount of drug could be maintained for almost a month by sustained release systems as proposed by [43]. These authors elaborated poly-lactic acid (RESOMER^®^ 202H) and PLGA (75:25) microspheres (MS) using oil-in-water (O/W) emulsion solvent-evaporation method. 20 to 45 µm poly-lactic acid MS prepared exhibit good physico-chemical properties and reduced burst effect (8.0 ± 1.3%). Brimonidine was released from MS imbebed in PBS (37 °C and pH = 7.4) by roughly zero-order kinetic. After a month, only 75% of the drug content was released. Administered in-vivo in the supraciliary space (adjacent to the drug’s site of action in the ciliary body using microneedles) of albino New-Zeland rabbits, these systems have demonstrated reduction of IOP for one month, but it would not be unreasonable to evaluate these systems for RGC protection. Upregulation of the glutamate transporters GLAST-1 and GLT-1 (another EAAT) has been shown to be the RGC neuroprotective mechanism behind neurturin [49]. No serious adverse effects were noticed, the eyes did not look inflamed and the animals did not show signs of pain, irritation or distress.

Finally, there must be mentioned that Ca^2+^ channel blocking has been proposed to be a mechanism that can be used to enhance RGC survival [50]. L-type Ca^2+^ channels play an important role in glutamate release from photoreceptors and bipolar cells. Dihydropyridines such as nimodipine and derivatives selectively blocks L-type Ca^2+^ channels and have demonstrated neuroprotection in cellular retinal ischemia/excitotoxicity models [51]. However, to date, results in our laboratory for RGC neuroprotection in a rat model of acute excitotoxicity has not shown positive results different from placebo. So, we agree with [52] that these statements are not reproduced in all studies, and it remains unclear whether this mechanism can exert a potential protective effect.

## 3. Neuroprotection

“Neuroprotective therapies” can be defined as pharmacological treatments focused on the relative preservation of neuronal survival, which is the reduction in the rate of neuronal loss over time [30]. These are plausible strategies for the treatment of various retinal dystrophies.

### 3.1. Neurotrophic Factors

The administration of neurotrophic factors is one of the most promising alternatives to enhance RGC survival. Over 2 decades ago, Faktorovitch et al. first proposed that neuro-growth factors, soluble small basic proteins for nervous system signaling, could be used to treat retinal degenerative diseases [53]. In general, a shortage of neurotrophins in the glaucomatous optic nerve has been implicated in RGC and photoreceptors loss [40]. Neurotrophic factors have the ability to promote the survival of neurons after optic nerve damage and to influence their growth [26]. Many of them (brain-derived neurotrophic factor, basic fibroblast growth factor, insulin-like growth factor 1, nerve growth factor and glial cell line-derived neurotrophic factor, to cite some of them) activate receptors that possess intrinsic tyrosine kinase activity resulting in upregulation of genes’ encoding antioxidant enzymes, and proteins involved in antiapoptotic pathways, plasticity, energy metabolism and ion homeostasis [54]. For this reason the retrograde axonal transport of neurotrophic factors synthesized in target structures has been specifically associated with RGC survival [26] and a shortage with poor prognosis. Nowadays, in addition, a novel indirect strategy that involves the administration of actives, such as growth hormones, that are able to increase over-expression of neurotrophic factors has been proposed [55].

Neurotrophic factors can be administrated intravitreally. However, they have short half-life values in the vitreous, which makes necessary the use of sustained delivery systems. Particular emphasis should be made on Human Glial cell line-derived neurotrophic factor (GDNF). This factor is a 20-kDa glycosylated protean homodimer belonging to the TGF-β-superfamily [26]. It is produced and release from Müller glia [5]. GDNF signals directly through the cell surface receptor, GFR- α, and indirectly through the transmembrane Ret receptor, tyrosine kinases like. At the cellular level, exogenous GDNF has demonstrated neuroprotective effect for RGC, promoting the survival of axotomized cells [56]. When it is intravitreally injected on rats, it improves the damaged RGC survival [57]. Beneficial effects of GDNF could be related to the increased levels of glutamate transporters shown after its administration, what increase the removal of glutamate from the synapse by GLAST-1 and protects the RGC against excitotoxicity [49,58]) and with the Müller cell proliferation observed in a pig model [59]. Because of the GDNF potency and mechanism of action, the concentration needed to provide neuroprotection is low; in vitro, the EC_50_ of GDNF that enhances dopaminergic neuron survival is 40 pg/mL [60].

Based on these observations, [61] elaborated by spontaneous emulsification method GDNF MP with an average diameter of approximately 10 mm that released in PBS at 37 °C on a labquake rotating shaker the drug over a total of 71 days with three stages. These authors observed a first release phase in the first day immediately following immersion of the spheres (20 ng or 59% of total release). This burst was followed by a 30-day plateau stage (1–2 ng/mg MP). During the final 40 days, 15 ng/mg was released (total cumulative release 35.4 ng/mg/71 days). MP were administered to an animal model suffering “early degeneration” of RGC (30% lost in the first 8 months) and “late degeneration (80% or RGC lost at 10 months). During the “early degeneration” after 8 months, RGC survival was 18.6%. After continuous delivery through 13 months, RGC densities were 2.9 times greater in treated eyes. However, the best part of their findings appears when the same GDNF MP were administered to what can be considered a “chronic” model of glaucoma. Hypertonic saline (1.9 M) was injected into the episcleral vein of the left eye in adult male Brown Norway rats while the right eye served as a normal control. Two weeks later, the injection procedure was repeated on a second episcleral vein on the opposite side of the same eye. The IOP elevation seen in this study was sustained up until the end of the experiment at the 10th week time point, eight weeks beyond the second hypertonic saline injection, that can be similar to the progression of glaucoma in human eye almost for what concerns IOP. Results were very satisfactory. The administration of 5 µl of a 10% GDNF microspheres suspension significantly increased RGC survival compared with either the administration of 5 µl of the 2% GDNF microspheres suspension, although this suspension resulted in significant preservation of RGCs compared with PBS treatment after seven weeks (total theorical amount 35.4 ng GDNF (obtained from [61])).

In an attempt to take it a step further, [62] proposed one of the most feasible therapeutic approaches. They elaborated PLGA microspheres (MS) using a novel S/O/W emulsion solvent evaporation technique including GDNF and vitamin E as an oily additive. The combination of several active substances with additives has been seen to be of optimal therapeutic strategy to achieve synergistic effects. Not only does vitamin E modulate release from microparticles allowing better-controlled release, but it also exerts a positive effect over RGC survival and optic nerve functionality. Protein was incorporated maintaining integrity into MS on solid state, protected from cavitation stress, in a high production yield. MS characterization results in spherical particles ranging from 19 to 26 µm (mean particle size 19.1 ± 9.4 μm) with a high number of pores in their surface. Particles in this range are suitable for administration as suspension through standard injection needles (27–34 G). The loading data obtained was 25.4 ± 2.9 ng GDNF/mg MS corresponding with an encapsulation efficiency (EE) of 27.8 ± 3.1%. The profile obtained when MS were suspended in PBS buffer alone and with different concentrations of BSA (0.1 and 1%) showed the typical triphasic shape of PLGA systems. The initial phase was characterized by a burst effect (protein release during the first 24 h of the release assay) followed by a short rapid release period, and a second long period of slow release. As injectable systems, these MS requires effective sterilization. Gamma-irradiation showed successful when MS where protected with dry ice, remaining the amount of protein release practically unchanged [63].

GDNF biological function preservation after microencapsulation and sterilization was first evaluated on retina cultures. The RGC survival percentage was higher than 70% for no sterilized MS addition in contrast to less than 28% when no sterilized blank microspheres were plated. After gamma-irradiation cell death showed similar results. In light of these good results, in vivo assays were carried out in adult male Brown Norway rats. Extremely low amounts of GDNF released from their MS (0.8 pg/day) achieved a therapeutic effect on the retina, protecting RGC in a glaucoma model (damage caused by increasing the IOP in a non-aggressive manner). RGC counting showed significant protection in animals administered MS. The average number of preserved RGC in treated rats was 51.6 ± 3.2 mm with 0.5% GDNF/Vit E microspheres (0.64 ng/eye) compared with 22.8 ± 3.2 mm using an equivalent amount of GDNF in a single dose. The percentage of axon survival was 72.68% with GDNF/Vit E microspheres compared to 28.96% with blank microspheres. No obvious side effects on the retinal integrity were noted. Recent studies confirmed this system to be safe. These MS did not show abnormalities during a six-month follow up after intravitreal administration on adult female New Zealand albino rabbits [32].

The insulin-related growth factors, proinsulin, insulin and insulin-like growth factor (IGF) I and II, regulate multiple processes in neural cells, including survival during development and in adult life [64]. In the eye, insulin and its precursor proinsulin delay photoreceptor cell death, preserves the structure and function of cones and rods, as well as their contacts with postsynaptic neurons [65]. PLGA proinsulin particles with a smooth surface and 10 to 32 μm in diameter, an adequate size for intravitreal injection, were elaborated by [53] from a W_1_/O/W_2_ emulsion using the solvent evaporation technique. Protein was not in solid state here, as in S/O/W emulsion techniques [17]. However, protein maintained enough conformation and integrity at least to exert therapeutic effects. The selected microsphere formulation, due to the needs of therapeutic concentration after 24 h, displays a high burst effect as well as continuous in vitro release of pronsulin for at least 45 days (release profile obtained in PBS, pH 7.4 and 37 °C). Notice that the authors find values on vitreous from undetectable to 203 pmol/g of total protein, with most values in the range of 16 to 32 pmol/g with undetectable levels in control eye and serum. This gives an idea of the low but sustained concentration that must be achieved in posterior segment of the eye and the possibilities that MP systems offer [35]. The neuroprotective effect was evaluated by ERG in mice in dark- and light-adapted conditions. ERG lines were better defined and greater in amplitude on treated eyes in comparison with control. The average b-mixed, b-cone, and oscillatory potential amplitudes were significantly higher in proinsuline-treated eyes versus control. While these authors did not specifically investigate the mechanism of action of proinsulin as a prosurvival molecule, it has been shown also to involve the activation of the PI3K pathway.

### 3.2. Antiapoptotic Compounds

An antiapoptotic compound is a substance for which biosynthesis and secretion do not occur in the mature nervous system and with an intrinsic ability of action in apoptosis pathways preventing their progress and promoting neuronal survival.

Tauroursodeoxycholic acid (TUDCA), the most important component of bear bile (*Ursus thibetanus* or *Selenarctos thibetanus*, also known as the Moon bear due to its coat markings [66]) has been shown to display cytoprotective and antiapoptotic effects in rodent models of retinal degeneration [14]. The mechanism of action is not fully understood though it seems to block apoptosis at various levels, including the alleviation of endoplasmic reticulum stress, the stimulation of the PI3K and MAPK (p38, ERK1/2) survival pathways and the blockade of Bax translocation to the mitochondria impeding subsequent cytochrome c release. These effects in the retina means less oxidative stress and inflammation overactivation what prevents microglia cascade pathway triggering and photoreceptor degeneration [8].

The authors of [14] developed, using an oil-in-water emulsion solvent evaporation method, novel 20 to 40 µm PLGA TUDCA MS (mean particle size 22.89 ± 0.04 μm), spherical in shape with a smooth surface in a high production yield (78.2 ± 2.1%). MS burst effect (drug released in the first 24 h) was low and represented only 4.45 ± 0.62% (0.55 ± 0.04 μg TUDCA/mg MSs) of the encapsulated drug. After that, two phases can be clearly distinguished. The first one had a slower release rate of 0.0368 μg TUDCA/mg MSs/day from day 1 to day 14, increasing to 0.2873 μg TUDCA/mg MSs/day from day 14 to day 28. After 28 days, at the end of the study, MS had released 40% of the content. After intravitreal MS administration (4 µL of a suspension of 5 mg TUDCA MS on 1.5 mL of PBS, pH = 7,4) in the right eye of homozygous P23H line albino rats (commonly accepted as a model of retinitis pigmentosa [67]) and age-matched Sprague-Dawley rats, on both groups, electrorretinograms responses were less deteriorated compared to left eyes responses where blank PLGA MS were injected as a control. As a result of the neuroprotection, higher a- and b-wave amplitudes were shown in the TUDCA-PLGA MS groups. Immunostaining with combinations of antibodies (anti-guinea pig IgG, anti-rabbit IgG and/or donkey anti-mouse IgG secondary antibodies at different dilutions, nuclear marker TO-PRO-3 iodide was also added) were used to evaluate the protective effect of the controlled delivery of TUDCA. To evaluate TUDCA controlled release ability to preserve retina, the degree of photoreceptor cells neurodegeneration was assessed. Few photoreceptors were found in the right P23H rat retinas compared to those observed in the right retinas of age-matched TUDCA-PLGA-MSs-treated animals. Apoptosis distribution was not homogenous throughout the retina and the number of preserved cells was higher in central areas of the retina with the maximum protection at the optic nerve level in the central retina. Protected photoreceptors maintain typical morphology and structure, with long axons, well-defined outer segments and typical pedicles containing numerous synaptic vesicles that surround well-structured synaptic ribbons. Cone photorreceptors in the negative control groups degenerate and cells were practically undistinguished. Finally, these authors demonstrate the preservation of synaptic contact between photoreceptor cells and second order neurons within the outer plexiform layer. A double immunostaining for α-PKC and Bassoon (a component of synaptic ribbons of both cone pedicles and rod spherules) evidenced the contact between the axon terminals of photoreceptor and bipolar cell dendrites. Not only was the dendritic arbor better conserved on PH23H rats, but also the contacts between photoreceptors and bipolar cells are similar to those observed in normal Sprague-Dawley retinas.

Other compounds have shown antiapoptotic effects in several animal models of ocular excitotoxicity. Between them, it is necessary to highlight I) apelin-36 and apelin- 17 involved on the activation of Akt and ERK1/2 signaling pathways required for neuronal survival and inhibition of apoptosis in the retina [68], II) cannabinoids via a mechanism involving the CB1 receptors, the PI3K/Akt and MEK/ERK1/2 signaling pathways [69], III) capsaicin, a transient receptor potential vanilloid type1 agonist that activates opioid receptors, calcitonin gene-related peptide receptor and the tachykinin NK1 receptor involved in the protective effect against the NMDA receptor induced neuronal death [70], IV) pituitary adenylate cyclase-activating polypeptide through phosphatidylcholine-specific PLC pathway and cAMP production [71], V) compounds acting on adenosine A_3_ receptor that attenuates the rise in calcium in RGC after activation of glutamate and P2X receptors protecting retinal cells, particularly RGC [72], VI) geranylgeranylacetone involved on the reduction in the activities of caspase-9 and -3, achieving protective results using a normal tension glaucoma mouse model which lacks GLAST [73], VII) CYM-5442 a known sphingosine 1-phosphate receptor agonist that administrated systemically in rats protected RGC from apoptotic death and preserve neuronal function after ET-1 induced RGC loss [19], VIII) adamantane derivatives such as memantine that blocks excessive activation of NMDARs without disrupting normal activity and has recently demonstrated significant preservation of RGCs density in a rodent ocular model of ocular hypertension when administrated in PLGA-polyethylene glycol (commonly PLGA-PEGylated) biodegradable nanoparticles [25], IX) tetramethylpyrazines, commonly known as TMPs, compounds able to block L-type voltage-gated calcium channels [7], X) tranylcypromine, a major lysine-specific demethylase 1 (LSD-1) and monoamine oxidase (MAO) inhibitor, that enhances expression of p38 MAPK and KEGG pathway genes [74], XI) melatonin, an autocrine or paracrine neuromodulator that regulates the local circadian physiology, that can be an effective antioxidant and antiapoptotic compound in the retina, acting as a direct and indirect free radical scavenger [75], XII) serotonin receptor (5-HT1A) agonists via the inhibition of cAMP-PKA signaling pathway that modulates GABAergic presynaptic activity [76], XIII) mTOR pathway stimulating APIs such as ciliary neurotrophic factors, lipopeptide N-fragment osteopontin mimic or lipopeptide phosphatase tension homologue inhibitors [77] or XIV) curcumin that modulates NMDA receptor subunit composition [78]. Finally, compounds with dual mechanism of action could be of special interest in multifactorial diseases as those related with RGC loss. In this sense, Mg acetyltaurate combines NMDARs’ inhibition and antioxidant effects [79] (Table 1).

Recently, in an interesting approach [80] prepared PLGA 50:50 MS by the Oil/Water emulsion solvent extraction-evaporation technique including dexamethasone, melatonin and CoQ10, three recognized neuroprotective agents that were able to provide simultaneous controlled co-delivery and maintain drug concentrations above the minimum effective level and the maximum safe concentration for, at least, more than a month. When multiloaded MSs formulation were administered and the in vivo neuroprotective effect over glaucoma induction in rats (by Morrison’s ocular hypertension model) evaluated, a significantly promotion of RGCs survival was found compared to administration of empty MSs. Special interest shows the fact that the administration of MSs loaded with a combination of the three drugs significantly promote RGCs survival compared to administration of empty MSs, what reveals that a combined therapy of some of those APIs could achieve synergistic results. The co-incorporation of different drugs into a single microcarrier can also reduce the amount of biomaterial (in this case PLGA) required for intraocular administration compared to equivalent dosing of single drug loaded formulations (here more than a half), which reduce the risk of PLGA associated retinal stress. Although several fixed combination therapies of antihypertensive drugs are currently in clinical practice, an equivalent neuroprotective combination therapy has not yet been clinically translated and that strategy may result in promising neuroprotective results for the treatment of multifactorial retinal diseases.

## 4. Retinal Structure Integrity

The structural complexity of the retina makes this tissue vulnerable to alterations from any sort of pathological injury. During retinal degeneration, retinal neurons are rewired while extracellular matrix (ECM) structural properties are changed. These changes alter matrix metalloproteinase (MMP) activity levels and influence cell-cell and cell-ECM interactions [81]. The authors of [82] demonstrated that MMP-2 and MMP-9 contribute to pathological remodeling of the inner limiting membrane and induce cell death. Their efforts to reduce MMP-mediated retinal damage with broad-spectrum inhibitors, such as GM6001, have produced encouraging results.

Recently, fragments of αβ crystalline released in a controlled way when fused with elastin like polypeptides have demonstrated prevention of retinal layers disruption, maintenance of transepithelial resistance and less disruption of tight junctions morphology in NaIO_3_ challenged mice [83]. The peptide construct generates spontaneous multivalent nanoparticles at physiological temperature. In spite of this, preclinical explorations of elastin like polypeptides support that their use in higher structures could enhance half-life of peptides or small proteins and achieve proper controlled release. Microparticles’ elaboration could be a good strategy here.

Also, connexins, transmembrane proteins serving as subunits of gap junction channels that forms two connexons, which allow rapid transport of ions or secondary messengers and small metabolites (up to 1 kDa) between connected cells within all tissues, play an important role in homeostasis too. Among all connexins, connexin43 is extensively located in the retina and its dysregulation has been observed in multiple neurodegenerative diseases. In fact, inside the retinal capillaries, endothelial cells are arranged very tightly between adjacent cells. Inferior up-regulation that results in excessive gap junction communication or uncontrolled opening of hemichannels that contribute to “center-surround” antagonism allow pro-inflammatory mediators activation, vascular permeability or interferences between intercellular transfer of ions and metabolites, ending in loss of RGC. Furthermore, it enables the release of apoptotic and necrotic signals from injured cells to the extracellular matrix where death signals can be passed to adjacent cells [37,84,85].

Thus, blocking the hemichannel opening but still keeping gap junction coupling should be explored as a potential neuroprotective tool. For this purpose, [37] try to demonstrate that hemichannel block could be potentially useful to limit blood retinal barrier disruption, inflammation, and lesion spread as well as RGC loss under retinal inflammatory and excitotoxic conditions. Based on previous studies that corroborate that structural mimetic connexin peptides, which are small peptides mimicking the sequence of connexin hemichannels, reduced cell death, optic nerve oedema, activation of astrocytes and microglia, and preserved vascular integrity, they elaborate connexin43 mimetic peptide PLGA micro (9.13 ± 0.37 μm, narrow size distribution) and nanoparticles (113.38 ± 0.74 nm) containing native connexin43 (drug loading 1.78 ± 0.05) using the double emulsion solvent evaporation method. Particles exhibited spherical structures and a relatively smooth surface. Zeta potential was neutral for MP and negative for nanoparticles at pH = 7.4. MP’s free diffusion through the vitreous via electrostatic repulsion between the particles and the negatively charged vitreous meshwork may be difficult, but it is much easier for nanoparticles. However, nanoparticles were almost completely eroded with all drug content release after 30 days in the usual release media (PBS, pH = 7.4 and 37 °C) while MP showed a slight pore formation on the particle surface and drug release up to 3 months in the classical profile, polymer degradation and drug diffusion through water-filled channels created during the erosion process. MP release was 28.6% of connexin43 following immersion (burst effect), 43.6% up to the 97th day and 27.8% during the final 14 days. Immediately following rats ischaemia–reperfusion, connexin43 MP diluted in 0.9% saline was intravitreally injected. The authors intended that MO administration reach a final peptide concentration of 20 μM assuming a vitreous volume of 50 μL. This system failed to rescue RGC. The most plausible explanation is the insufficient initial connexin43 release from MP during the immediate acute stage after injury. In this case, the results suggest that the RGC rescue effect may be more attributable to the initial burst of connexin43 than the continuous slow release, suggesting that the sooner connexin43 hemichannels are blocked after ischaemia the better. Despite nanoparticles showing the most promising results in the acute model employed by these authors, it must be taken into account that neurodegenerative diseases characterized by slow and chronic progression and death of RGC significantly increases with time [38]. The biological study evaluated here suited to rapid RGC degeneration and treatments were tested in the short term. This fact, linked with the reduced injection frequency required due to the sustained release effect that renders the MP formulation vs. the bolus administration and the reduced drug concentration needed in vitreous, but expected to be high that the dose administered somelike insignificant, spotlight that MP formulations has significant potential in the treatment of chronic retinal excitotoxicity and may still provide long-term RGC protection.

## 5. Conclusions

Excitotoxicity seems to play a critical role in ocular neurodegeneration. Excess glutamate mediated RGC death is the principal cause of cell loss. Uncontrolled glutamate in the synapsis have significant implications in the pathogenesis of neurodegenerative disorders. At present, there are at least four mechanisms that have provided enough evidence of ocular protection where drugs could act. Acting over excess glutamate and/or its impacts, avoid the induced apoptosis underlying NMDA over-activation or the loss of retinal integrity and the use of neuroprotectants (Figure 2). The exploitation of approaches of controlled release systems enhance the pharmacokinetic and pharmacodynamic activity of drugs. Furthermore, these systems are even more interesting when, as here, sustained but minimal drug concentrations are required. In particular, microparticles are secure, can maintain therapeutic drug concentrations in the eye for prolonged periods and make intimate contact improving drug biodisponibility. These are numerous and good enough reasons to further study this ophthalmic systems to design formulations that have, in animal models, demonstrated no toxicity but a statistical reduction of RGC degeneration, loss of functionality and morphology changes. According to the promising results reported (Table 2), new investigations will focus possibly much attention on microparticulate formulations which can be expected to open the field to new alternatives for doctors, as required by patients.

## Figures and Tables

**Figure 1 pharmaceutics-12-00094-f001:**
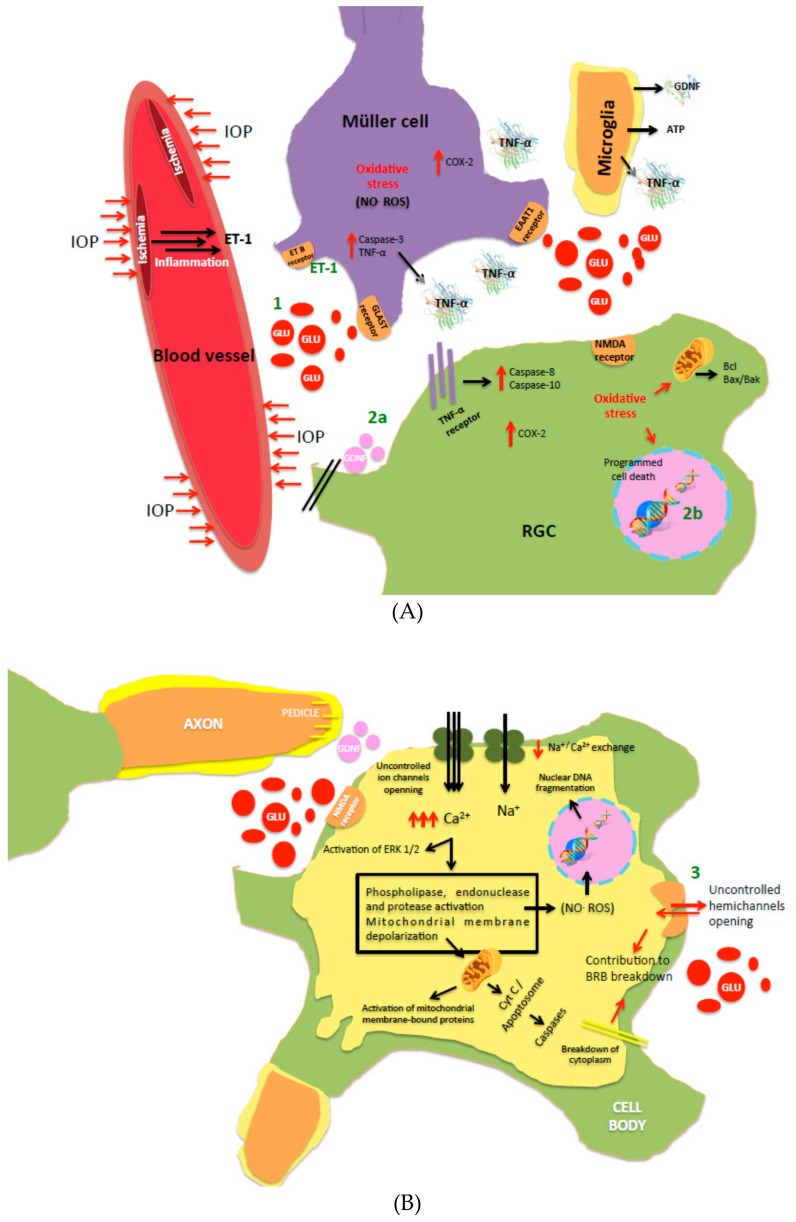
Schematic representation of (**A**) a synapse where intraocular pressure (IOP) and inflammation are causing optic nerve damage through excitotoxicity and the events that occur in retinal ganglion cells (RGC), Müller cells and microglia; (**B**) intracellular mechanisms activated in RGC after accumulation of excess glutamate in the synapse that lead to apoptosis. A detailed study can be found in [18] and [5]. Possible interventions to reduce excitotoxicity are indicated; 1) represents intervention over excess glutamate, 2a) administration of neurotrophic factors, 2b) administration of antiapoptotic compounds and 3) maintenance of retinal structure integrity.

**Figure 2 pharmaceutics-12-00094-f002:**
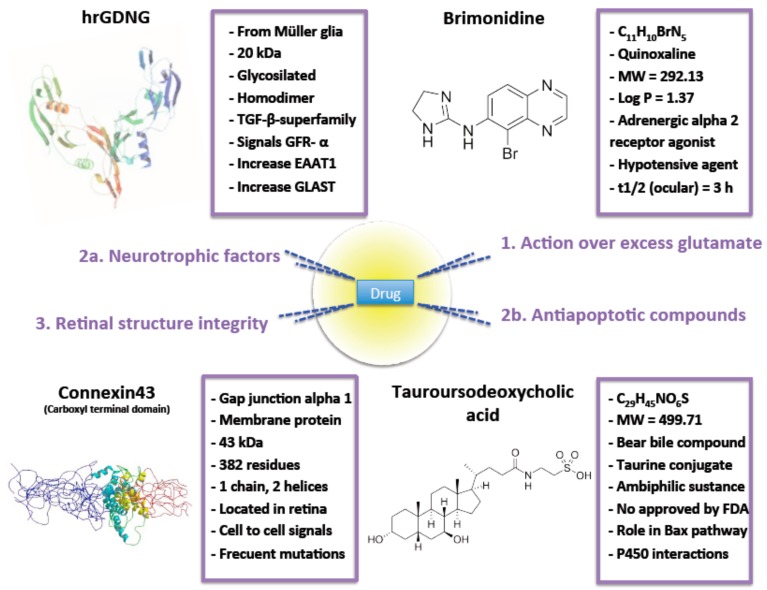
Drugs incorporated into microparticles to treat excitotoxicity. For each compounds the principal characteristics are highlighted.

**Table 1 pharmaceutics-12-00094-t001:** Compounds evaluated in retina cells for their antiapoptotic activities. The pathway and/or mechanism involved/proposed is included.

Compounds	Pathway/Mechanism Involved	Bibliography
Apelin-36 and apelin- 17	Akt and ERK1/2 signaling pathways.	[68]
Cannabinoids	CB1 receptors, PI3K/Akt and MEK/ERK1/2 signaling pathways.	[69]
Capsaicin	Opioid, calcitonin gene-related peptide and tachykinin NK1 receptor.	[70]
Pituitary adenylate cyclase-activating polypeptide	Phosphatidylcholine-specific PLC pathway and cAMP production.	[71]
Adenosine A_3_ receptor agonists	Attenuates the rise in calcium in RGC.	[72]
Geranylgeranylacetone	Reduction in the activities of caspase-9 and caspase-3.	[73]
CYM-5442	Sphingosine 1-phosphate receptor agonism.	[19]
Adamantane derivatives	Blockage of NMDARs excessive overactivation	[25]
Tetramethylpyrazines	Blockage of L-type voltaje-gated Ca^2+^ channels.	[7]
Tranylcypromine	P38 MAPK and KEGG pathway genes expression.	[74]
Dual compounds (e.g., Mg acetyltaurate)	NMDAR inhibition + antioxidant effect.	[79]
Melatonin	Direct and indirect free radical scavenger.	[75]
5-HT1A agonists	Inhibition of cAMP-PKA signaling pathway.	[76]
Ciliary neurotrophic factors, lipopeptide *N*-fragment osteopontin mimic, lipopeptide phosphatase tension homologue inhibitors	mTOR pathway stimulation.	[77]
Curcumin	Modulation of NMDA receptor subunits composition.	[78]

**Table 2 pharmaceutics-12-00094-t002:** Alternatives for ocular neuroprotection where microparticles (MP) have been evaluated. The compound tested and its properties as well as the most relevant results obtained are shown.

Neuroprotection based on	Compound	Properties	Encapsulation	Observations	Bibliography
**Intervention over excess glutamate**	Brimonidine	See Figure 2	Poly-lactic acid (RESOMER^®^ 202H) MS	Particle size between 20 to 45 µm. Reduced burst effect. After a month, only 75% of the drug was released. Reduction of IOP after a month. No serious adverse effects noticed and eyes did not look inflamed and the animals did not show signs of pain, irritation or distress. RGC protective activity was evaluated.	[43]
**Neuroprotective therapies**					
Neurotrophic factors	GDNF	See Figure 2	PLGA (50:50) MS	Particle size ≈ 20 µm. Drug loading ≈ 25 ng/mg. EE ≈ 28%. RGC survival in-vitro > 70%. 50% higher preservation of RGC in vivo compared to the same dose of GDNF administered in bolus. No side effects observed on retina.	[32,62,63]
Antiapoptotic compounds	TUDCA	See Figure 2	PLGA (50:50) MS	Particle size ≈ 20 µm. Spherical MS. High production yield. Low burst effect. Significant photoreceptor’s survival. Well-preserved contact between photoreceptor cells and second order neurons.	[14]
	Dexamethasone (DX) Melatonin (Mel) Coenzyme Q10 (CoQ10)	Antiapoptotic Antioxidant Anti-inflammatory	PLGA (50:50) MS	Particle size ≈ 24 µm. Spherical MS. Production yield ≈ 72% EE ≈ 78% DX; 62% Mel; 96% CoQ10 Low burst effect and triphasic release. Neuroprotection—high RGC density—in the Morrison’s model of ocular hypertension; whole retina density measures demonstrated that MS administration preserved RGC to a comparable extent as naïve retinas. Multidrug MS demonstrated less side effects than the same amount of drug administered in single-drug loaded MS.	[80]
**Retinal structure intregrity**	Connexin43 mimetic peptide	See Figure 2	PLGA (50:50) MP	Particle size ≈ 9 µm and narrow distribution. Spherical morphology. Smooth surface. Neutral zeta potential. MP release drug in sustained release more than 3 months. No enough drug released after a day to exert effective protection maybe due to rapid RGC death after ischemia lesion.	[37]

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
