# Peer review of "Retinal Cell Protection in Ocular Excitotoxicity Diseases. Possible Alternatives Offered by Microparticulate Drug Delivery Systems and Future Prospects"

_pharmaceutics, 2020, doi:10.3390/pharmaceutics12020094_

Round 1

Reviewer 1 Report

The manuscript written well and explained more information on the excitotoxicity effect on retinal protection. However, address very minor comments.

Review mainly explained on the effect of excitotoxicity on glaucoma. Include the role of the glutamate effect on other retinal damaged diseases.

What is the effect of various routes of administration on the retinal damage not explained?

Effect of implants as delivery system not discussed.

Author Response

Dear reviewer, 

Thanks for your comments,

As a balance between all reviewers' comments, introduction is not going to be further extended, but clarified as you proposed. 

The importance of increased glutamate levels (excitotoxicity) in other degenerative ocular diseases is now remarked and cited.

The risks of (frequent) intravitreal administration is included and cited.

The possibility of microparticles' behaviour as an implant has been included and cited.

Sincerely,

Dr. Javier Rodríguez

Reviewer 2 Report

This review summarized the microparticles-mediated treatment strategy to protect retinal cell.

Although this manuscript is well written, major revision is needed.

More detail comments are listed below.

1) Figure 1 is too small and is not informative. Furthermore, Figure 1 is almost the same as Figure 2. Reviewer could not know which pictures in Figure 1 indicate the microparticulate drug delivery systems and future prospects.

2) Authors should add the description regarding the ocular trafficking and distribution of microparticles.

3) In lines 287-288, authors described ‘The combination of a neurotrophic factor and an antioxidant agent…….’ Authors should explain why the combination is more effective. Reviewer could not know whether the descriptions among lines 293-300 are a only GDNF treatment or the cotreatment of GDNF with antioxidant.

4) In lines 350-352, authors described ‘Recent studies confirmed this system to be safe’ and ‘These MS showed any abnormalities during a six……….’ These sentences are contrary.

5) What are PEGylated biodegradable nanoparticles in line 438?

6) In Table 1, the information regarding microparticles should be added.

7) The text regarding Figure 3 is not found.

8) Is Connexin 43 in figure 3 the mimetic peptide? If so, authors should add the description regarding the sequence and function of the peptide.

9) Can authors combine Figure 3 with Table 2?

Author Response

Dear reviewer,

Thanks for your comments.

As a result, we have discussed and decided to delete Fig. 1. not to mislead the reader. As a consequence, Fig.2 and Fig. 3 are now Fig. 1 and Fig. 2.

Also, the description regarding the ocular trafficking and distribution of microparticles has been included and cited.

L287-300 has been clarified to explain why the contribution is more effective. 

L354 should say "didn’t show abnormalities".

PLGA-PEGylated biodegradable nanoparticles are PLGA-polyethylene glycol nanoparticles. Now is defined properly. 

Table 1 only exemplify compounds that have demonstrated antiapoptotic activity and might be included in MP formulations.

Reference to Fig.2 and Tab. 2 has been inserted correctly.

Function of connexin 43 is included in Fig. 2, but we cannot reach the point of introduce the sequence. However, "frequent mutations" was selected as a relevant data. 

Combination of figure 2 and table 2 was our first option but it results in a large table that seem difficult to read so, definitely, the idea was discarded.

Sincerely,

Dr. Javier Rodríguez

Reviewer 3 Report

The article itself discusses an interesting topic and the pharmaceutics is being discussed in an adequately thorough manner, however there are some important concerns with the overall structure. The primary concern here is the excessive depth given both in the introduction as well as while introducing each approach or intervention. As this is a review in pharmaceutics, more emphasis was expected on the evidence seen with specific delivery systems and approaches and less on the mechanism of the diseases or of the therapeutics or on how specific studies are carried out. The reviewer appreciates that these are important to explain at some level, however the level used in this article is substantially more than necessary.

Not many specific comments can be offered as the core content has been well discussed. Apart from shortening the background descriptors throughout the manuscript, the following should be looked at:

There are numerous minor but frequent typos occurring throughout the text Figure 1 – some text is difficult to read even at 300% magnification, and the different aspects have not been well-described in the figure legend. The figure also seems unnecessary as it does not add much to the reader’s understanding of the review. Figure 2 – “ischemia” is difficult to read. A lot is going on in the figure but it is not being well explained in the legend. Table 1 – the different compounds being evaluated would be of more interest to a pharmaceutical audience if specific delivery systems that they have been incorporated in were also listed. Table 2 – in column 1, starting the different headings with a number is distracting (e.g. “4.Retinal structure integrity”. The reviewer understands why this has been done, but it compromises the tidiness of the article.

Author Response

Dear reviewer,

Thanks for you comments.

Regarding the complain of excessive depth in some parts of the MS, we need to balance the comments of all reviewers and some of them emphasis that we should explain in more detail, for example, the state of art and methodology of some papers. Finally, we have decided not to extend the content itself cause we understand your concern, but clarify some parts to improve quality.

Figure 1 has been deleted from the MS due to the overall negative comments.

Figure 2 (and its legend) has been modified as proposed. 

Table 1 only exemplify compounds that have demonstrated antiapoptotic activity and might be included in MP formulations.

Table 2 has been modified as proposed.

Sincerely,

Dr. Javier Rodríguez Villanueva

Round 2

Reviewer 2 Report

Authors answered to all my comments point by point.
I think that this review will be acceptable.

Author Response

Dear reviewer,

Thanks for your comments.

Sincerely,

Dr. Javier Rodríguez Villanueva

Reviewer 3 Report

Both Figure 1 and 2 are showing a lot of boxes with question marks. Please check the formatting here.

Author Response

Dear reviewer, 

Thanks for your comment. We have (re)uploaded the MS in PDF.

Sincerely,

Dr. Javier Rodríguez Villanueva